# Effect of *Moringa oleifera* Leaf Extract on Excision Wound Infections in Rats: Antioxidant, Antimicrobial, and Gene Expression Analysis

**DOI:** 10.3390/molecules27144481

**Published:** 2022-07-13

**Authors:** Abdullah A. Al-Ghanayem, Mohammed Sanad Alhussaini, Mohammed Asad, Babu Joseph

**Affiliations:** Department of Clinical Laboratory Science, College of Applied Medical Sciences, Shaqra University, Shaqra 11961, Saudi Arabia; alghanayem@su.edu.sa (A.A.A.-G.); malhussaini@su.edu.sa (M.S.A.); masad@su.edu.sa (M.A.)

**Keywords:** *Moringa oleifera*, antioxidant, antimicrobial, wound healing activity

## Abstract

The present study investigated the wound healing activity of *Moringa oleifera* leaf extract on an infected excision wound model in rats. Infection was induced using methicillin-resistant *Staphylococcus aureus* (MRSA) or *Pseudomonas aeruginosa.* An investigation was also done to study the effect of *Moringa* extract on the vascular endothelial growth factor (VEGF) and transforming growth factor-beta 1 (TGF-β1) gene expression in vitro using human keratinocytes (HaCaT). The methanol extract of *M. oleifera* leaves was analyzed for the presence of phytochemicals by LCMS. The antimicrobial activity of the extract was also determined. Wound contraction, days for epithelization, antioxidant enzyme activities, epidermal height, angiogenesis, and collagen deposition were studied. *M. oleifera* showed an antimicrobial effect and significantly improved wound contraction, reduced epithelization period, increased antioxidant enzymes activity, and reduced capillary density. Effect of the extract was less in wounds infected with *P. aeruginosa* when compared to MRSA. The VEGF and TGF-β1 gene expression was increased by *M. oleifera*.

## 1. Introduction

Wound healing is the body’s response to injury and helps to restore the skin structure through various mechanisms that include inflammatory response and proliferative activity by the involvement of different cells [1]. Infection in wounds by multidrug-resistant bacteria aggravates skin damage and reduces the efficacy of antibiotics, resulting in treatment failure and reoccurrence of infections [2]. The use of medicinal plant products having wound healing potential with antimicrobial effects benefits mankind clinically and economically [3].

*Moringa oleifera* Lam (Family: *Moringaceae*) is widely used as food and traditional medicine. It contains several nutraceuticals and is reported for various pharmacological activities including antimicrobial, antioxidant, anticancer, and antidiabetic properties [4]. Alkaloids, polyphenols, phenolic acids, a range of flavonoids, and glusinolates present in this plant have been reported for different biological activities [5]. *M. oleifera* has been reported for wound healing effects [6,7] and antimicrobial properties against different bacterial pathogens [8,9].

The incidence of bacterial infections in wounds is increasing worldwide [10]. Adaptation strategies, survival mechanisms, and the development of antibiotic resistance mechanisms by bacterial pathogens are making the treatment of infections more difficult and expensive [11]. Methicillin-resistant *Staphylococcus aureus* (MRSA) and *Pseudomonas aeruginosa* (*P. aeruginosa*) are common bacterial pathogens causing nosocomial infections, especially in the wound, skin, and soft tissues [12]. In addition to this, both these bacterial pathogens make a biofilm on wounds leading to reduced healing, extensive damage to the wounded tissues, and the development of antibiotic resistance [13]. Clinical management of such infections on wounds is challenging, and indiscriminate and prolonged use of anti-infectives leads to serious therapeutic problems [14]. 

The management of wound healing and wound infection mainly involves treatment with antibiotics and analgesics. Despite concerns about the effectiveness of antibiotic treatments, they are still the most widely used therapeutic agents in the treatment of wounds. In fact, it is believed that patients with wounds receive more antibiotics compared to other aged-matched and gender-matched patients [15]. Apart from this, resistance to antibiotics used in treating wound infection is becoming a serious health issue [16]. Other newer techniques for wound management, such as photomodulation, skin substitutes, and external tissue expanders, have been used with varying success but these treatments are expensive [17]. Therefore, different phytochemicals and novel bioactive compounds are under investigation to treat wound infections to control multi-drug resistant microbial pathogens [18]. We have previously reported the effect of *M. oleifera* extract on diabetic wound infection in rats wherein a significant wound healing effect was observed [19]. In this study, the wound healing potential of *M. oleifera* extract on excision wound model inoculated with either MRSA or *P. aeruginosa* in normal rats was determined along with its cytotoxicity and effect on expression of the vascular endothelial growth factor (VEGF) and transforming growth factor-beta 1 (TGF-β1) genes on human keratinocyte (HaCaT) cell lines under in vitro conditions.

## 2. Results

### 2.1. Phytochemical Analysis

Different classes of constituents such as alkaloids, flavonoids, steroids, tannins, and polyphenols were revealed by preliminary phytochemical analysis. The LCMS analysis of the crude extract revealed a large number of phytoconstituents (Table 1). The total ion chromatogram (positive mode), and total ion chromatogram (negative mode) are given as Figure 1 and Figure 2 respectively.

### 2.2. Physicochemical Properties of the Extract

The physicochemical characteristics of the emulsifying ointment revealed that it was very stable, homogenous, diffused through an agar medium, and had good spreadability at room temperature. It was green in color with a bitter taste.

### 2.3. Antibacterial Activity

The prepared formulation of the extract inhibited both bacterial pathogens. However, MRSA showed more susceptibility compared to *P. aeruginosa.* The extract showed a minimum inhibitory concentration (MIC) for MRSA at 0.512 (±0.03) mg/mL and the minimum bactericidal concentration (MBC) was 1.024 (±0.04) mg/mL. The MIC for *P. aeruginosa* was 1.024 (±0.04) mg/mL and MBC was 2.048 (±0.01) mg/mL. The values given above are mean ± SD. 

### 2.4. Wound Healing and Epithelization

To determine the wound healing effect, the wound area was measured on days 4, 8, 12, 16, and 20. The wound area in animals treated with the extract (20%) decreased promptly compared to the control (Figure 3).

MRSA infected wound healing: MRSA infection on wounds significantly reduced the healing process (Figure 4). The contraction of wounds was decreased when compared to uninfected wounds. Application of *M. oleifera* extract augmented wound healing, which was dose dependent. The higher concentration (20% *w/w*) showed more wound contraction compared to the lower concentration (10% *w/w*). However, the wound contraction with extract was lower when compared to standard antibiotic mupirocin. The epithelization period also was delayed in MRSA-infected wounds (Figure 5). The treatment with extract also reduced the epithelization period in MRSA-infected wounds. Mupirocin treatment showed a better effect in reducing the epithelization period compared to the extract. 

The antioxidant enzymes SOD and catalase were decreased in the wounded tissue in the MRSA-infected wound. There was a significant increase (*p* < 0.001) in the enzyme activity after the application of *M. oleifera* extract. Mupirocin also induced an increase in antioxidant enzyme activity, though the effect was lower compared to the *M. oleifera* extract (Figure 6).

MRSA infection suppressed wound healing that was indicated by a reduction in the epidermal height, number of capillaries, and inflammatory cells with less collagen in the wounded area. Application of *M. oleifera* extract increased epidermal height, angiogenesis, and collagen deposition when compared with infection control. Inflammatory cells were also reduced in the treated wounds. Nevertheless, reduced inflammatory cells, epithelial height, and collagen deposition were noticed after mupirocin application (Figure 7 and Figure 8). The microbial load was significantly reduced after treatment with the extract and antibiotics. In the infected animals, the bacterial count on the 20th day was around 81.5 × 10^4^ CFU/g of the tissue. In extract-treated groups, it was reduced to about 27.3 × 10^4^ CFU/g with 20% of the extract and to 32.7 × 10^4^ CFU/g after the application of the 10% of the extract. It was lowest in the antibiotic-treated animals, where the CFU was around 19 × 10^4^ CFU/g.

*P. aeruginosa infected wound healing: P. aeruginosa* induced severe infection on the wounds that manifested as fluid oozing out of the wounded tissues leading to a delay in wound healing. The period of epithelization was longer, and the wound contraction in animals with infection was delayed when compared to wounded animals without infection. The lower concentration (10% *w/w*) of *M. oleifera* extract did not produce much wound healing action on infected models whereas the higher concentration (20% *w/w*) showed wound healing activity after 8 days. Animals treated with gentamicin showed significant wound contraction from the 4th day onwards (Figure 9). The epithelization of the wound was delayed in infected wounds compared to normal wounds. There was a delay in wound healing and epithelization after *P. aeruginosa* infection when compared with MRSA (Figure 10).

Antioxidant enzymes; SOD and catalase activities were decreased in *P. aeruginosa* infected wounds. However, the application of the extract (20 % *w/w*) increased these enzyme activities, whereas a lower concentration (10% *w/w*) had a limited effect on antioxidant enzymes. Application of gentamicin on *P. aeruginosa*-infected wounds promoted wound healing and increased antioxidant enzymes (Figure 11).

Histological studies revealed less regeneration of the epidermal layer in *P. aeruginosa*-infected control animals. The well-developed epidermal layer was evident in gentamicin treated wounds, followed by *M. oleifera* extract (20% *w/w*) in treated wounds, and a lesser effect was observed at a low concentration of extract (10% *w/w*). A reduction in collagen deposition, broken epithelial layer, and reduced number of capillaries with more inflammatory cells were observed in the infected skin of control animals. Application of *M. oleifera* (20% *w/w*) or gentamicin to wounds infected with *P. aeruginosa* augmented epidermal regeneration, collagen formation, and increased capillary formation (Figure 12 and Figure 13). The microbial load in *P. aeruginosa* treated control was around 58 × 10^5^ CFU/g, while in the 10% extract treated group it was 42.4 × 10^5^ CFU/g. The effect was more in the 20% extract-treated group, where it was reduced to about 15.7 × 10^5^ CFU/g, and in the gentamicin treated animals it was around 13 × 10^4^ CFU/g tissue.

### 2.5. Skin Irritation Study

The application of extract on the skin did not produce any erythema or inflammation even after 72 h. 

### 2.6. Cytotoxicity and Gene Expression Studies on HaCaT Cells

The *M. oleifera* leaf extract was safe, and no cytotoxicity on HaCaT cells was observed up to a concentration of 1000 µg/mL (Figure 14). However, lower concentrations of extract increased cell viability. *M. oleifera* increased VEGF gene expression and TGF-β1 gene expression in the HaCaT cell line (Figure 15).

## 3. Discussion

The present study focused on the antibacterial and wound healing effects of *M. oleifera* extract. Pathogens commonly causing infections in wounds were selected for the present study. This study evaluated the antibacterial effect of the extract on the multidrug-resistant strains of the microorganisms along with the promotion of wound healing. *M. oleifera* extract has been reported earlier for wound healing activity [27]; however, the present study confirms the inhibition of pathogenic bacteria on wounds as well as its wound healing potential. *M. oleifera* is widely consumed as food, and earlier reports indicate that oral administration did not inhibit MRSA infection [28]. Therefore, this research was conducted by local application of methanol extract on the infected wounds. Further, antioxidant activity and histological evaluations were carried out to understand the wound healing potential of the extract. The cytotoxicity of the extract and its effect on the expression of VEGF and TGF-β1 genes were evaluated using HaCaT cells. The study was a continuation of our report on the effect of *M. oleifera* extract on infected wounds in diabetic rats [19]. 

Phytochemical analysis revealed the presence of different components such as flavonoids, alkaloids, tannins and phenols. Numerous bioactive components have been reported in *M. oleifera* including phenolic acids and flavonoids [29]. The leaves of *Moringa* are rich in these components compared to bark and seeds [30]. Therefore, the leaves were used to prepare the extract for the study. Most of the studies on *Moringa* have been reported in aqueous extracts, whereas a methanolic extract was shown to have more bioactive constituents [31]. Further, identification different of components were carried out by the LCMS analysis.

Formulation of the extract as an ointment was based on the British pharmacopoeia method using paraffin oil, emulsifying wax and soft paraffin [32]. The preparation was confirmed for diffusion on agar during antibacterial activity. The selection of bacterial isolates was based on the available literature, common opportunistic pathogens, and pathogens delaying wound healing. Wound infections with MRSA and *P. aeruginosa* have been shown to prolong healing time and induce adversarial postoperative outcomes [13]. The virulence factors and surface proteins of these pathogens delay the process of wound healing [13]. Delayed wound healing with bacterial infections results in high treatment costs and increased morbidity [33]. Prevention of bacterial infections is an important priority in cutaneous wound management [34]. In this study, MRSA or *P. aeruginosa* was selected to represent two wide groups of Gram-positive and Gram-negative bacteria to establish the wide spectrum activity of the extract. Additionally, these pathogens are responsible for hospital-acquired infections. Preventing infection on wounds and treatment remains challenging due to the development of resistance and the tendency to form biofilm on wounds that protects pathogens from host defense mechanisms [35]. To overcome antibiotic resistance development in bacterial pathogens, natural components and novel phytochemicals with high potency to heal wounds and antimicrobial properties are being explored [36,37].

Different concentrations of *M. oleifera* showed potential wound healing activity. There was no sign of skin irritation. All parameters tested in the present study indicate the healing process of the wound. The contraction of the wound was measured every 4 days to understand the wound healing progress and epithelization was recorded to determine complete wound healing [38]. The application of *M. oleifera* improved the healing process in animals infected with MRSA or *P. aeruginosa*. However, the wound contraction was quick in MRSA-infected wounds than the *P. aeruginosa*-infected wounds. Stress mediated via oxidative species was determined by measuring antioxidant enzyme activities. These enzymes support wound healing by removing the free radicals that are known to impair the wound healing [39]. Several constituents with antioxidant effects are known to be present in *M. oleifera* including quercetin and kaempferol [40]. An increase in activity of antioxidant enzymes after treatment with *M. oleifera* extract leads to free radical scavenging that improves the healing of infected wounds. 

Macroscopic investigations on the healing of the infected wound were further observed by microscopic observation in histological examinations. Epidermal regeneration, number of capillaries and presence of inflammatory cells was determined by H & E stain, and collagen deposition was determined using Masson’s trichrome stain. Epithelial height shows clear development of epithelium, and capillary density indicates angiogenesis confirming the wound healing. The presence of collagen was observed in treated tissues, which improves tensile strength. Inflammatory cells in the wound area indicate the initial phase of healing and their abundance 20 days after wounding suggests incomplete wound healing [41]. The increased epidermal heights and capillary density, with less inflammatory cells in treated wounds, indicated increased wound healing. 

HaCaT cells are widely used to evaluate wound healing responses in the skin [42]. No cytotoxic effect was observed with *M. oleifera* extract in the MTT assay, further confirming its safety on the skin. The expression of VEGF and TGF-β1 were studied in presence of extract on HaCaT cells. Vascular endothelial growth factor (VEGF) is a pro-healing cytokine well-known for endothelial cell generation, promoting cell migration, chemotaxis, and vascular permeability [43]. Another important cytokine, transforming growth factor-beta 1 (TGF-β1), is involved in stimulating angiogenesis, the proliferation of fibroblast, and remodeling of new extracellular matrix collagen synthesis and deposition [44]. In the present study, expression of both these genes was increased in HaCaT cells with the presence of extract in the medium, with the assumption that it had improved wound healing. *Moringa* has been reported to release other cytokines that are involved in inflammation and immune reactions [45]. Apart from the expression of VEGF and TGF-β1, other cytokines may also be involved in improving wound healing. 

The results of the current study on infected wounds in normal rats support our findings on the effect of *M. oleifera* extract on the infected wounds in diabetic rats [19]. In our earlier study, we analyzed the chemical constituents of *M. oleifera* by GC-MS to show the presence of volatile constituents. In the present study, a detailed LC-MS analysis was done to determine all the constituents. The effect observed on the healing of wounds in normal rats in the present study was similar to that observed in diabetic animals. However, *M. oleifera* extract showed relatively greater effect in healing wounds in normal rats compared to that observed in diabetic animals in our earlier study. The difference in healing of wounds induced by MRSA or *P. aeruginosa* in both normal and diabetic animals was similar. 

## 4. Materials and Methods

*Chemicals:* All chemicals used were of analytical grade purchased from different suppliers. For qualitative test of phytochemicals, reagents that include different reagents such as Dragendroff’s reagents, Benedict’s reagent, and Meyer’s reagent, and other chemicals, were procured from LobaChemie (India) and SD Fine (India). Different reagents and chemicals used for the determination of antioxidant enzyme levels were from LobaChemie (India) and SD Fine (India). Media for microbiological assays were purchased from Hi-Media (India) while the constituents for preparation of emulsifying base were purchased from a local pharmacy store. The spectrophotometer used for determination of antioxidant enzymes was from Shimadzu (Japan).

*Microorganisms:* Methicillin-resistant *Staphylococcus aureus* (ATCC 43300) and *P. aeruginosa* (ATCC 27853) were used.

*Extract preparation:* Fresh leaves from *M. oleifera* plant were collected during the month of January 2022 from the institutional campus. The identification was carried out by a botanist in the institute through a voucher specimen (SU/CAMS/07/2021) preserved in the institute. A methanolic extract of the leaves was prepared by hot extraction using a soxhlet apparatus [46]. Briefly, powdered leaves were packed in the central chamber of soxhlet apparatus with methanol as the extracting solvent in the flask below. The extraction temperature was kept at the boiling point of methanol (65 °C) and the extraction was continued until a drop of the solvent from the siphon did not leave any residue when evaporated. A yield of the 19.52% *w/w* of the leaves was obtained.

*Preliminary and chromatographic analysis:* Preliminary phytochemical investigation was done using standard tests for qualitative determination of different classes of compounds such as alkaloids, flavonoids, tannins, and steroids [47]. LCMS was also carried out. The instrument details are given in Table 2.

*Preparation of extract ointment:* The ointment was prepared by a fusion method in two different concentrations (10% *w/w* and 20% *w/w*) using an emulsifying base [48]. Physicochemical, stability and diffusion properties of the formulation were evaluated and standardized [49]. Emulsifying base without the extract was used as a control in untreated animals. 

*Antimicrobial activity:* The minimum inhibitory concentration (MIC) was determined using Mullen Hilton broth. The minimum bactericidal concentration (MBC) for MRSA was determined using mannitol salt agar and cetrimide agar was used for *P. aeruginosa* [2].

*Animals:* Adult albino in-bred Wistar rats were used. Animals weighing 215 to 245 g of either sex aged between 4 months to 4.5 months were selected. Animals were provided with water and normal diet ad libitum. Standard protocols were followed according to the university guidelines. Test animals were handled with all precautionary measures to prevent transmission of MRSA or *P. aeruginosa*, and infected animals were kept in a separate room. The Ethical Research Committee of Shaqra University approved the research methodology (Approval number -53/11600). All measurements were done by investigators who were blinded to the treatment.

*Wound healing activity:* For anesthetizing the animals, a ketamine (91 mg) and xylazine (9.1 mg) cocktail was administered intraperitoneally at a dose of 1 mL/kg [50]. The back region of the animals was shaved and an area of 500 mm^2^ was marked. The skin was excised to full thickness. MRSA or *P. aeruginosa* was inoculated into the wounds, and animals without inoculation served as control. For each pathogen, five group of animals consisting of twelve animals were used. For six animals in each group, the wound contraction was determined during the treatment and the wound tissues were collected on the 20th day for determination of antioxidant enzyme activities, histological studies, and for the determination of microbial load, while the other six animals were used to evaluate the epithelization period. In both the study group of pathogens, the first group animals served as control without inoculation of the pathogen with the application of base only. In the other four groups, wounds were inoculated with 30 µL of bacterial culture (10^8^ CFU/mL) of either MRSA or *P. aeruginosa*. The second group of animals was treated with an ointment base, the third group with 10% *w/w* and the fourth group with 20% *w/w* extract incorporated ointment every day. The last group received antibiotics, which were mupirocin (2% *w/w*) in the case of MRSA-infected animals and gentamicin (0.1% *w/w*) for *P. aeruginosa*-infected animals. Wound area was measured every 4th day by tracing on a transparent sheet to determine wound contraction. On the twentieth day, six animals were anesthetized and the tissues were collected for determining superoxide dismutase (SOD) activity [51], catalase activity [52] and histological parameters. The skin sections for histological examination were stained with hematoxylin and eosin (H & E) or Masson’s trichrome stain. Another six animals were observed for complete epithelization and complete healing process. The sections were observed under a magnification of 100× using a Leica DM 2500 LED connected to a camera (DFC 295). The pictures were acquired using Leica LAS EZ software (Leica, Germany).

*Skin irritation test:* Prior to the study, skin on the dorsal side of rats was shaved and the extract was applied (500 mm^2^). The area was covered using adhesive tape. The skin was observed at different time intervals at 1, 24, 48 and 72 h and untreated skin served as control.

*In-vitro study on HaCaT cells*:

*Cytotoxic effect:* Am MTT assay was performed using HaCaT cells (National Centre for Cell Sciences, Pune, India). Dulbecco’s Modified Eagle Medium (DMEM) supplemented with 1.5 g/L glucose, 10% fetal bovine serum (FBS) and 1% antibiotic solution was prepared in 96 well plate and 1000 cells/ well were inoculated then incubated at 37 °C with 5% CO_2_. Different concentrations of extract (1–1000 µg/mL) were added into wells followed by 24 h incubation. MTT solution (250 µg/mL) was added and incubated for 2 h. Culture supernatant was removed and 100 µL DMSO was added to dissolve the cell layer matrix. The results were recorded using a plate reader.

*VEGF and TGF-β1 gene expression:* HaCaT cell lines grown in a 96 well plate for 24 h in DMEM were used. The medium was supplemented with 1.5 g/L glucose and incubated for 24 h with extracts at a nontoxic concentration. The wells without extracts served as controls. RNA was isolated from the cells using the trizol method (Thermo Scientific). Denatured agarose gel (1.5% *w/v*) was used to check the integrity of the isolated RNA. The primers used are given Table 3. The cDNA was synthesized using a standard protocol. The RT-PCR cycle was an initial denaturation at 95 °C for 3 min, denaturation at 95 °C for 30 s, annealing at 60 °C for 30 s, and final extension at 72 °C for 55 s, with a total run of 35 cycles. 

*Statistical analysis:* All results show mean ± SEM in the footnotes. One-way ANOVA with Tukey’s post-test was done using SPSS.

## 5. Conclusions

The multiple mechanisms of *M. oleifera* extract, such as antibacterial, antioxidant, and proliferative actions, supported healing in infected wounds when applied locally. However, this extract was less effective in *P. aeruginosa*-infected wounds, showing that the highly infected wound needed more days for epithelization and healing. The different constituents present in the extract are reported for their biological activities. Hence, it can be assumed that no single component is responsible for the healing of infected wounds, as observed in the present study. Further investigations of these components might help in determining their contribution to the observed effect. Generally, the results support the traditional use of *M. oleifera* for healing wounds with its antibacterial and antioxidant activities, along with the expression of important genes such as VEGF and TGF-β1. The extract showed a better wound healing activity on MRSA-infected wounds compared to *P. aeruginosa*. Further investigations on different bioactive components are needed for preparing a better formulation in controlling infections on wounds. 

## Figures and Tables

**Figure 1 molecules-27-04481-f001:**
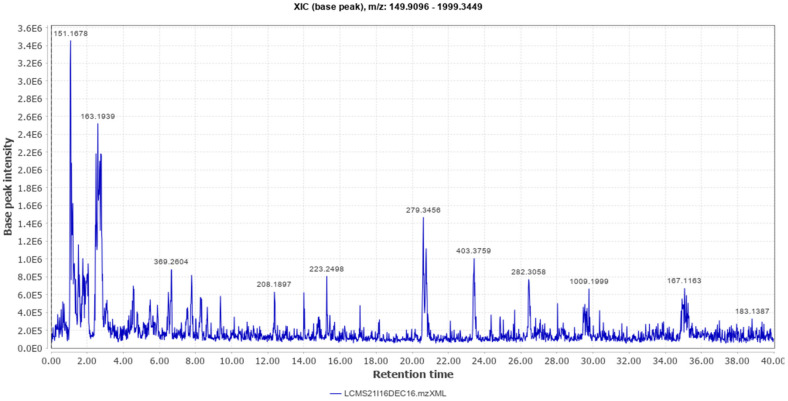
Total Ion Chromatogram (Positive Mode).

**Figure 2 molecules-27-04481-f002:**
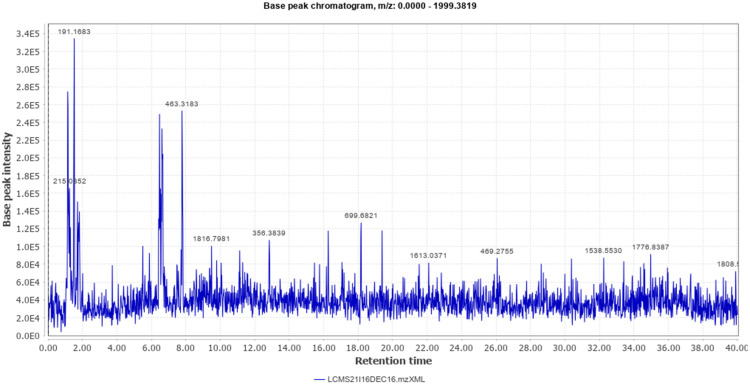
Total Ion Chromatogram (Negative Mode).

**Figure 3 molecules-27-04481-f003:**
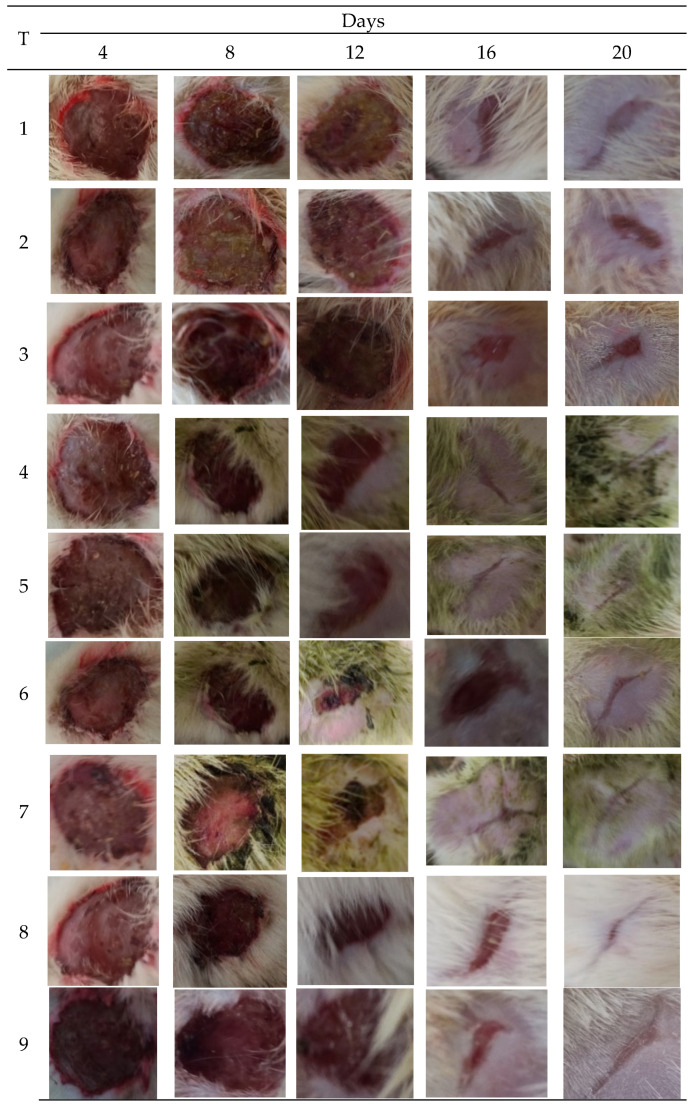
Images of wounds infected with bacterial pathogens treated with different concentrations of *M. oleifera* extract, along with control. Treatments: 1. Ointment base; 2. MRSA-infected wound treated with ointment base; 3. *P. aeruginosa* infected wound treated with ointment base; 4. MRSA-infected wound treated with 10% extract; 5. MRSA-infected wound treated with 20% extract; 6. *P. aeruginosa*-infected wound treated with 10% extract; 7. *P. aeruginosa*-infected wound treated with 20% extract; 8. Infected with MRSA and treated with mupirocin (positive control); 9. Infected with *P. aeruginosa* and treated with gentamicin (positive control).

**Figure 4 molecules-27-04481-f004:**
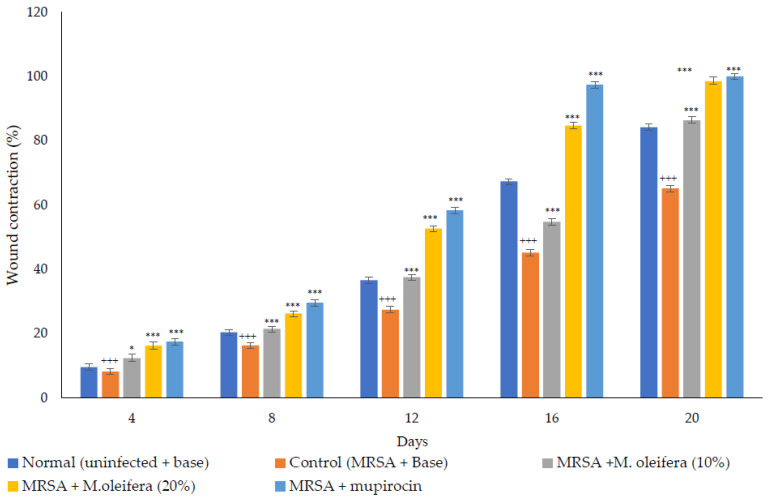
*M*. *oleifera* extract ointment preparation on the percentage of wound contraction in MRSA-infected wounds. All values are mean ± SEM, n = 6, * *p* < 0.05 when compared to control; *** *p* < 0.001 when compared to control; +++ *p* < 0.01 when compared to normal.

**Figure 5 molecules-27-04481-f005:**
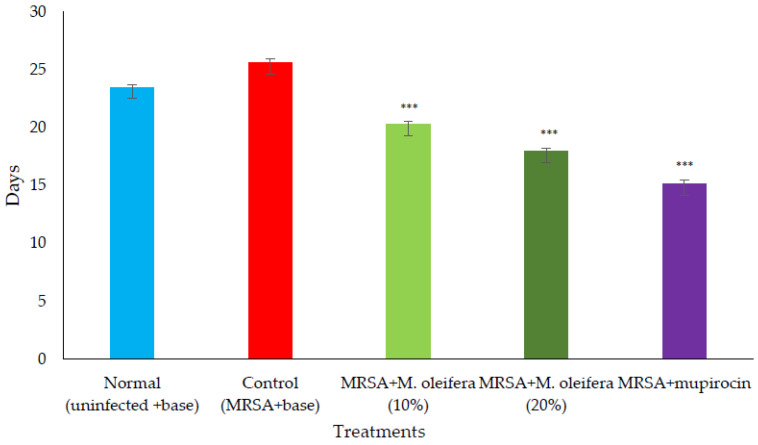
Epithelization period in *MRSA*-infected wounds in rats. All values are mean ± SEM, n = 6, *** *p* < 0.001 when compared to control.

**Figure 6 molecules-27-04481-f006:**
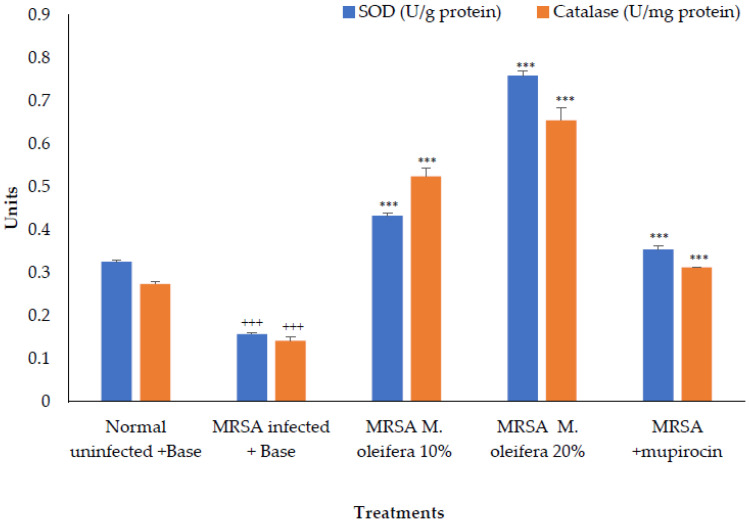
Effect on antioxidant enzymes in MRSA-infected wound tissues of rats. All values are mean ± SEM, n = 6, +++ *p* < 0.01, *** *p* < 0.001 when compared to control.

**Figure 7 molecules-27-04481-f007:**
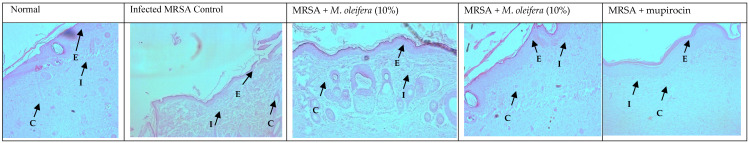
Skin histology of MRSA-infected wound (stained with H & E) after different treatments showing epidermis (E), capillaries (C), and inflammatory cells (I).

**Figure 8 molecules-27-04481-f008:**
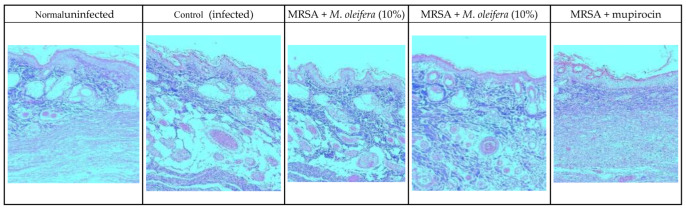
Skin histology of MRSA-infected wound (stained with Masson’s trichrome stain) after different treatments. The blue color indicates collagen.

**Figure 9 molecules-27-04481-f009:**
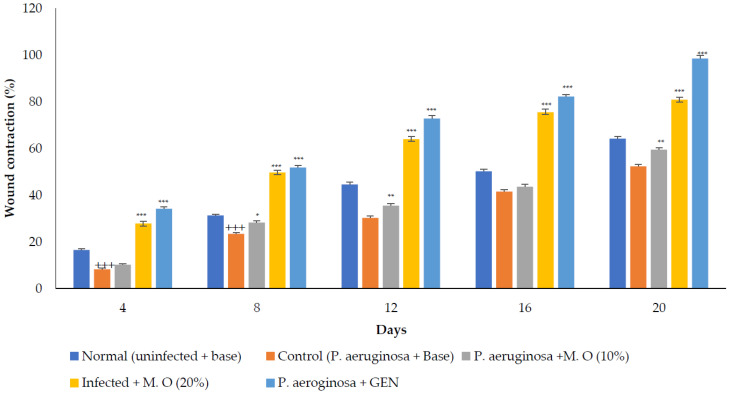
*M*. *oleifera* extract ointment preparation on the percentage of wound contraction in *P. aeruginosa*-infected wounds. All values are mean ± SEM, n = 6, +++ *p* < 0.001 when compared to normal. * *p* < 0.05, ** *p* < 0.01, *** *p* < 0.001 when compared to control.

**Figure 10 molecules-27-04481-f010:**
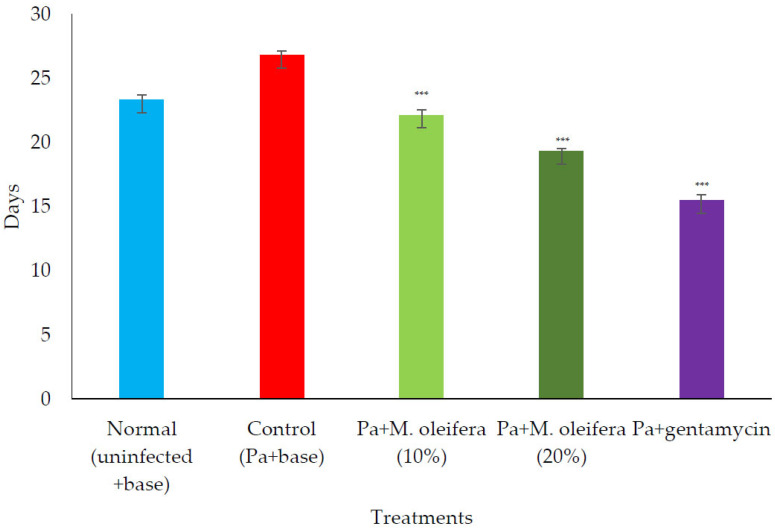
Effect on the period of epithelization in *P. aeruginosa* (Pa)-infected wounds in rats. All values are mean ± SEM, n = 6, *** *p* < 0.001 when compared to control (infected).

**Figure 11 molecules-27-04481-f011:**
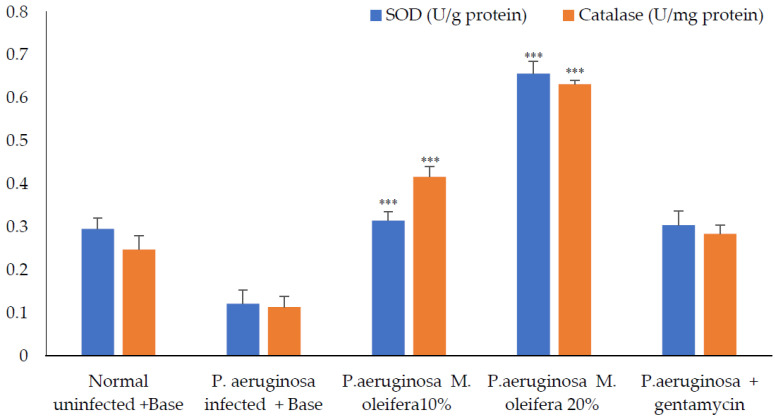
Antioxidant enzymes activity after *P. aeruginosa* infection. All values are mean ± SEM, n = 6, *** *p* < 0. 01 when compared to control (infected).

**Figure 12 molecules-27-04481-f012:**
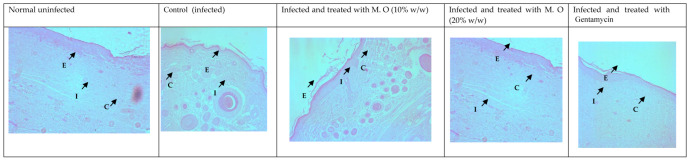
Skin histology of *P. aeruginosa*-infected wound (stained with H & E) after different treatments showing epidermis (E), capillaries (C), and inflammatory cells (I).

**Figure 13 molecules-27-04481-f013:**
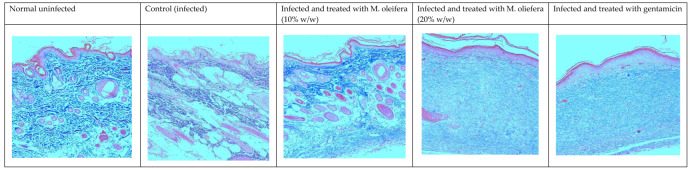
Skin histology of *P. aureginosa*-infected wound (stained with Masson’s trichrome stain) after different treatments. The blue color indicates collagen.

**Figure 14 molecules-27-04481-f014:**
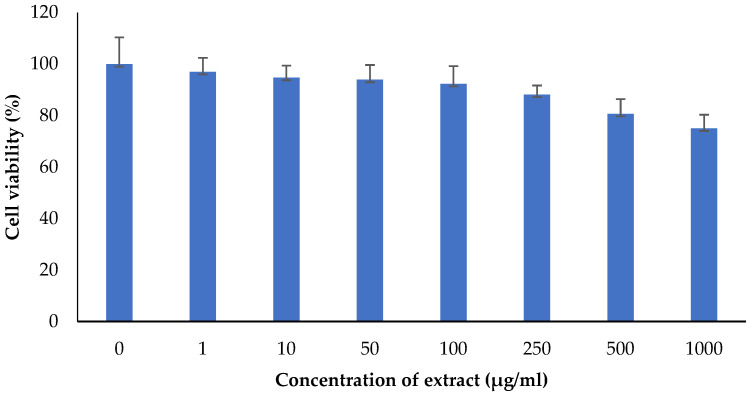
Effect of *M. oleifera* extract on the cell viability of human keratinocytes (HaCaT) cells. All values are mean ± SEM, n = 6.

**Figure 15 molecules-27-04481-f015:**
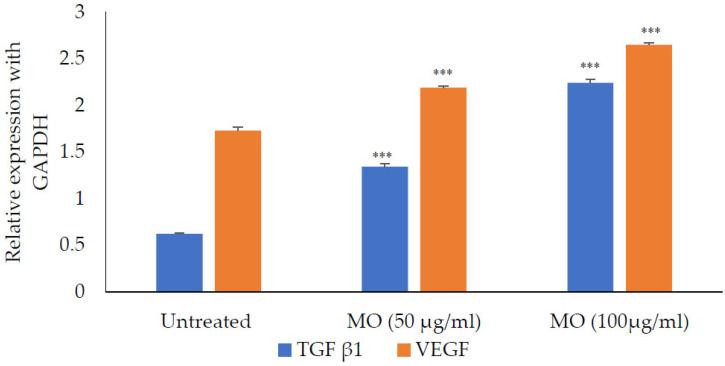
Expression of TFG-β1 and VEGF in the presence of *M*. *oleifera* (MO) extract. All values are mean ± SEM, n = 6, *** *p* < 0.001 when compared to untreated cells.

**Table 1 molecules-27-04481-t001:** List of compounds revealed by LC-MS analysis of *Moringa* extract.

No.	Retention Time (RT) (min)	Formula	Calculated Mass (Da)	Theoretical Mass (Da)	Mass Error (ppm)	MSE Fragmentation	Identification	Ref.
1	1.6	C_7_H_6_O_4_	151.17	154.0266	3.2	153.0215[M − H]^−^, 135.0211[M-H-H2O]^−^, 89.0340[M-H-H2O-HCOOH]^−^	3,4-Dihydroxy-benzoic acid	[20]
2	1.8	C_7_H_12_O_6_	191.1683	192.0634	−1.10	191.0542[M − H]^−^, 173.0432[M-H-H2O]^−^, 145.0516[M-H-HCOOH]^−^, 137.0232[M-H-3H2O]^−^, 127.0401[M-H-H2O-HCOOH]^−^	Quinic acid	[21]
3	3.1	C_9_H_8_O_3_	163.19	164.0474	0.3	165.0544[M + H]^+^, 147.0444[M + H-H2O]^+^, 119.0483[M + H-HCOOH]^+^	o-Coumaric acid	[22]
4	6.6	C_17_H_20_O_9_	369.26	368.1107	−1.5	367.1029[M − H]^−^, 336.0902[M-H-OCH3 ]^−^, 295.1124[M-H-4H2O]^−^, 243.0591[M-H-CH3-C6H5O2]^−^, 189.0549[M-H-CH3-C9H7O3 ]^−^, 178.0346[M-H-C8H13O5]	Methyl-3-caffeoylquinate	[23]
5	7.7	C_21_H_20_O_12_	463.31	464.0955	−3.4	463.0866[M − H]^−^, 318.0758[M-H-2H2O-C6H5O2]^−^, 178.0513[M-H-C15H9O6]^−^, 159.0379[M-H-Glu-C6H4O3]^−^	Isoquercetin	[24]
6	7.9	C_21_H_20_O_12_	464.0955	464.0949	−1.3	465.1022[M + H]^+^, 285.0485[M + H-Glu]^+^, 231.0678[M + H-Glu-3H2O]^+^, 149.0150[M + H-Glu-C7H4O3]^+^, 152.0154[M + H-Glu-C8H5O2]^+^	Hyperoside	[25]
7	12.4	C_10_H_12_O_2_	164.0837	164.0831	−2.8	209.1118[M + HCOO]^−^, 122.0453[M-H-C3H5]^−^, 105.0495[M-H-OCH3-C2H3]^−^	Eugenol	[24]
8	13.1	C_16_H_18_O_9_	356.38	354.0951	0.1	353.0878[M − H]^−^, 253.1035[M-H-3H2O-HCOOH]^−^, 190.0182[M-H-3H2O-C6H5O2]^−^, 144.0302[M-H-H2O-C7H11O6]^−^, 125.0251[M-H-H2O-HCOOH-C9H8O3]^−^	Chlorogenic acid	[23]
9	15.1	C_12_H_16_O_4_	224.1038	224.1049	−4.8	223.0965[M − H]^−^, 205.1027[M-H-H2O]^−^, 135.0421[M-H-C4H8O2]^−^, 123.0964[M-H-C4H4O3]^−^, 87.0295[M-H-C8H8O2]	3-Butylidene-4,5,6,7 -tetrahydro-6,7-dihydroxy-1(3H)-isobenzofuranone	[26]
10	15.3	C_12_H_14_O_4_	222.0883	222.0892	−3.9	221.0811[M − H]^−^, 160.0546[M-H-OC2H5]^−^, 119.0282[M-H-C3H5O2 -C2H5]^−^,	Diethyl phthalate	[23]
11	20.9	C_18_H_30_O_2_	278.2237	278.2246	−3.0	277.2165[M − H]^−^, 182.1234[M-H-C7H11]^−^, 168.1230[M-H-C8H13]^−^, 110.0795[M-H-C11H17-H2O]^−^	Linolenic acid	[21]
12	23.7	C_20_H_26_O_9_	410.1573	410.1577	−1.0	409.1504[M − H]^−^, 336.0817[M-H-C4H9O]^−^, 251.1394[M-H-2H2O-C7H6O2]^−^, 202.0639[M-H-C3H7-C9H7O3]^−^, 134.0437[M-H-C12H19O7]^−^	5-O-Caffeoylquinic acid butyl ester	[20]
13	26.9	C_18_H_34_O_2_	282.2558	282.2559	−0.4	283.2631[M + H]^+^, 97.1020[M + H-C5H11-C6H11O2]^+^, 86.1024[M + H-C12H21O2]^+^, 72.0876[M + H-C13H23O2]^+^	Oleic acid	[21]
14	39.2	C_9_H_16_O_4_	188.1045	188.1049	2.1	187.0965[M − H]^−^, 141.1105[M-H-HCOOH]^−^, 123.0957[M-H-H2O-HCOOH]^−^, 112.0644[M-H-H2O-C3H5O]^−^	Azelaic acid	[20]

**Table 2 molecules-27-04481-t002:** Instrument details for LC-MS analysis.

LC Instrument	XEVO-TQD#QCA1232
Column	SUNFIRE C18, 250 × 2.1, 2.6 μm
**HPLC Conditions**	
A%	0.0 H_2_O
B%	5.0 ACN
C%	0.0 MeOH
D%	95.0 0.1% Formic Acid in water
Flow (mL/min)	1.500
Stop Time (min)	5.0
Column Temperature (°C)	30.0
Min Pressure (Bar)	0.0
Max Pressure (Bar)	300.0

**Table 3 molecules-27-04481-t003:** Primer sequences of target genes.

Gene	Forward	Reverse
**GAPDH**	5′CGGAGTCAACGGATTTGGTCGTAT3′	5′AGTCTTCTCCATGGTGGTGAAGAC3′
**TGF-β1**	5′CTTCTCCACCAACTACTGCTTC3′	5′GGGTCCCAGGCAGAAGTT3′
**VEGF**	5′CTGGCCTGCAGACATCAAAGTGAG3′	5′CTTCCCGTTCTCAGCTCCACAAAC3′

## Data Availability

Data is contained within the article.

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
