# Peer review of "Effect of Moringa oleifera Leaf Extract on Excision Wound Infections in Rats: Antioxidant, Antimicrobial, and Gene Expression Analysis"

_molecules, 2022, doi:10.3390/molecules27144481_

Round 1

Reviewer 1 Report

Please refer the attachment

Author Response

  1. Change the keywords.

It is done now as per the reviewer’s suggestion.

  1. Please provide LCMS chromatogram.

It is given now as figure 1 and figure 2.

  1. How to author determine the compounds in the extracts. Need to explain in the manuscript.

The compounds were determined by preliminary phytochemical analysis that involves qualitative tests to detect the presence of phyto-constituents. This was mentioned in the manuscript. These are simple test that involves color/precipitation reactions. A reference for the same is now given.

  1. Section 2.3 – What is the SD value for MIC and MBC?

The SEM is now given.

  1. Some statement in the manuscript without the reference. Please add the citation.

The manuscript was thoroughly checked and eight more references have been added for different statements where references were not given earlier.

  1. 4 – Material and Methods.
    1. List the instruments.

The instruments used have been mentioned under LC-MS method and gene expression assay. For antioxidant enzyme activity, spectrophotometer was used. It is now mentioned.

  1. List down the chemicals.

The number of chemicals and reagents were used for qualitative assay and antioxidant enzyme activities were large. It is not feasible to provide the list of all chemicals and their supplier. However, a statement about the suppliers is now given under chemicals

  1. Who is identify the plant material & what is the voucher specimen?

A botanist identified the plant. This is now mentioned in the manuscript along with the voucher specimen number.

  1. Please state the detail extract preparation.

The soxhlet method used for extraction is a standard procedure most widely used in phytochemistry. A detailed explanation of the procedure should include complete description of the soxhlet apparatus with a diagram, which is not necessary as it is available in several phytochemistry and pharmacognosy books. Hence, a brief explanation about the solvent, temperature and how extraction was done has been given now.

  1. Explain the chromatographic analysis. Type of column, detector, solvent system, flow rate, etc.

It is now given as Table 2.

  1. This study involves the animals. Please provide the animal ethic

It is already given under the heading Animals as “The Ethical Research Committee of Shaqra University approved the research methodology (Approval number -53/11600).

Reviewer 2 Report

In their manuscript, the authors wounded then infected wistar rats with two pathogenic bacteria, namely, Staphylococcus aureus (MRSA) and Pseudomonas aeruginosa. After this, they assessed the healing effect of Moringa oleifera leaf extract on the wound created. They also evaluated the antioxidant and antimicrobial activities of this extract. Briefly, they showed that, as compared to wounds inoculated with P. aeruginosa, wounds contaminated with S. aureus healed faster when treated with M. oleifera leaf extract. They demonstrated that this extract possesses an antioxidant and antimicrobial effect. Taken together, I congratulate the authors for conducting this work and getting such amazing results.  But before recommending this manuscript for publication, I would like the authors to consider my following comments.

Abstract

Line 10 Please replace "investigates" with "investigated"

Introduction

I suggest the authors add a paragraph presenting the management techniques that are available for treating infection induced by S. aureus and P. aeruginosa, and if possible their limitations.

Line 77. Rather than stating "data not shown", why not put them as the supplementary results and state them here? I think if the authors want to be trusted by the readers, they should do this.

I think that the manuscript contains a lot of figures which according to me should be merged in a single manuscript and labelled as "A" and "B". For instance, Figures 2 and 7 present data about wound contraction when rats are infected by S. aureus and P. aeruginosa respectively. The authors can simply these two figures together and label them A  and B. Same comment goes with figures 3 and 8, figures 4 and 9, figures 6 and 11.

Please be aware that in your manuscript, figure 5 is not shown anywhere (you have figures 1, 2, 3, 4,........., 6, 7, 8, 9, 10, 11, 12, and 13)

Author Response

Abstract

Line 10 Please replace "investigates" with "investigated"

Correction made

Introduction

I suggest the authors add a paragraph presenting the management techniques that are available for treating infection induced by S. aureus and P. aeruginosa, and if possible their limitations.

It is given now as per the reviewer’s suggestion

Line 77. Rather than stating "data not shown", why not put them as the supplementary results and state them here? I think if the authors want to be trusted by the readers, they should do this.

The MIC and MBC values along with SEM are now given in the text. Hence, ‘data not shown’ is now deleted.  In the skin irritation study, there is no data as such, it is just a picture of normal skin after application of extract ointment. The words ‘data not shown’ is now deleted

I think that the manuscript contains a lot of figures which according to me should be merged in a single manuscript and labelled as "A" and "B". For instance, Figures 2 and 7 present data about wound contraction when rats are infected by S. aureus and P. aeruginosa respectively. The authors can simply these two figures together and label them A  and B. Same comment goes with figures 3 and 8, figures 4 and 9, figures 6 and 11.

We agree with the reviewers that figures can be merged and labeled as A and B. We tried it too. The figures became very big and confusing. Furthermore, it was difficult to distinguish between the effect on MRSA infection and P. aeruginosa infection. Hence, it was not done. However, if the reviewer still feels that figures have to be merged and the readers will be able to distinguish the difference between the MRSA and P. aeruginosa, we will do it.

Please be aware that in your manuscript, figure 5 is not shown anywhere (you have figures 1, 2, 3, 4,........., 6, 7, 8, 9, 10, 11, 12, and 13).

We have shown Figure 5 and Figure 6 in the same page. Figure 5 is shown as “Figure 5. Skin histology of MRSA infected wound (stained with H & E) after different treatments showing epidermis (E), capillaries (C), and inflammatory cells” This is now Figure 7 because chromatogram was added as per the suggestions of the other reviewer.

Round 2

Reviewer 1 Report

After careful reading this manuscript and cross-checking with other sources. I found a similar publication had been published in MDPI (Pharmaceuticals). Title: Moringa oleifera Leaf Extract Promotes Healing of Infected Wounds in Diabetic Rats: Evidence of Antimicrobial, Antioxidant and Proliferative Properties. Pharmaceuticals (2022) 15(5). https://doi.org/10.3390/ph15050528. Please give the explaination regarding this matter. 

Other comments are

1) Suggestion of Keywords: Moringa oleifera, Antioxidant, Antimicrobial, Wound healing activity

2) I have doubt regarding SD value because it is too low.  

Author Response

After careful reading this manuscript and cross-checking with other sources. I found a similar publication had been published in MDPI (Pharmaceuticals). Title: Moringa oleifera Leaf Extract Promotes Healing of Infected Wounds in Diabetic Rats: Evidence of Antimicrobial, Antioxidant and Proliferative Properties. Pharmaceuticals (2022) 15(5). https://doi.org/10.3390/ph15050528. Please give an explanation regarding this matter. 

We are thankful to the reviewer for referring to our earlier paper. We carried out both studies simultaneously. We have now cited the reference of our earlier published paper in the introduction as” We have previously reported the effect of M. oleifera extract on diabetic wound infection in rats wherein a significant wound healing effect was observed”

And in the discussion in two places as

 “The study was a continuation of our report on the effect of M. oleifera extract on infected wounds in diabetic rats”

 “The results of the current study on the infected wounds in normal rats support our findings on the effect of M. oleifera extract on the infected wounds in diabetic rats [19]. In our earlier study, we had analyzed the chemical constituents of M. oleifera by GC-MS to show the presence of volatile constituents. In the present study, a detailed LC-MS analysis was done in order to determine all the constituents. The effect observed on the healing of wounds in normal rats in the present study was similar to that observed in diabetic animals. However, M. oleifera extract showed relatively greater effect in healing wounds in normal rats compared to that observed in diabetic animals in our earlier study. The difference in healing of wounds induced by MRSA or P. aeruginosa in both normal and diabetic animals were similar”

Other comments are

  • Suggestion of Keywords: Moringa oleifera, Antioxidant, Antimicrobial, Wound healing activity

We have replaced the key words as per the suggestion of the reviewer

  • I have doubt regarding SD value because it is too low.  

We appreciate the comments of the reviewer. We had mentioned SEM values in our revision. As per the reviewer’s suggestions, the SD values are now given.
